# Optimal Design and Competitive Spans of Timber Floor Joists Based on Multi-Parametric MINLP Optimization

**DOI:** 10.3390/ma15093217

**Published:** 2022-04-29

**Authors:** Primož Jelušič, Stojan Kravanja

**Affiliations:** Faculty of Civil Engineering, Transportation Engineering and Architecture, University of Maribor, Smetanova ulica 17, 2000 Maribor, Slovenia; stojan.kravanja@um.si

**Keywords:** structural optimization, cost optimization, discrete optimization, mixed-integer non-linear programming, MINLP, timber floor joists

## Abstract

This study investigates the optimization of the design of timber floor joists, taking into account the self-manufacturing costs and the discrete sizes of the structure. This non-linear and discrete class of optimization problem was solved with the multi-parametric mixed-integer non-linear programming (MINLP). An MINLP optimization model was developed. In the model, an accurate objective function of the material and labor costs of the structure was subjected to design, strength, vibration and deflection (in)equality constraints, defined according to Eurocode regulations. The optimal design of timber floor joists was investigated for different floor systems, different materials (sawn wood and glulam), different load sharing systems, different vertical imposed loads, different spans, and different alternatives of discrete cross-sections. For the above parameters, 380 individual MINLP optimizations were performed. Based on the results obtained, a recommended optimal design for timber floor joists was developed. Engineers can select from the recommendations the optimal design system for a given imposed load and span of the structure. Economically suitable spans for timber floor joists structures were found. The current knowledge of competitive spans for timber floor joists is extended based on cost optimization and Eurocode standards.

## 1. Introduction

Timber floor joists consist of parallel timber beams and a structural sheathing, joined together by shear connectors (see Figure 1). Floors made of timber, along with those made of reinforced concrete and steel-concrete composites, are among the most common forms of construction used in the modern building industry [1]. The advantages of timber floors are: use of a natural building material; use of a sustainable building material [2]; low greenhouse gas emissions [3,4]; agreeable interior climate [5]; diverse architectural design options; good ratio between self-weight and load-carrying capacity [6]; fast-dry form of construction; and no scaffolding or props required for erection. Timber floor joists are easily adapted to various levels of loading, fire resistance and sound insulation [7]. This floor system has low self-weight and is highly flexible in terms of fitting out. The disadvantages of floor joists are that considerable structural depth is required and a limited range of solid timber sections is available [8]. The vibration responses of a timber floors have been assessed experimentally [9,10] and numerically [11], and it has been concluded that vibration serviceability of timber floors is increasingly relevant due to their increased long-span applications [12]. With increased floor span, sawn timber beams are replaced with glued laminated timber beams, and secondary timber beams are provided. Based on thermo-mechanical analysis and experimental testing of timber floor systems, design methods for calculating fire resistance have been developed [13].

Nowadays, structural efficiency, economy and environmental behavior are the dominant criteria in the structural design of floor systems [14]. In order to reduce manufacturing costs or mass, various floor structures have often been optimized. While the optimizations of concrete slabs and composite floors have been frequently performed and widely reported, such investigations on timber floors have been very rarely performed according to literature. Among the references in the last five years in the field of structural optimization of concrete slabs is the optimization of large-span prestressed concrete slabs by Kaveh et al. [15] using probabilistic particle swarm, cost optimization of reinforced concrete flat slabs by Aldwaik and Adeli [16] using robust neural dynamics model, design optimization of reinforced concrete slabs by Basha and Latha [17] using the genetic algorithms, optimization of reinforced concrete slab by Stochino and Lopez Gayarre [18] using simulated annealing, the design optimization of reinforced concrete slabs by Suryavanshi and Akhtar [19] using various metaheuristic optimization algorithms, the optimization of reinforced concrete slabs according to Eurocode 2-EC2 by Fedghouche [20] using the generalized reduced gradient algorithm, the cost optimization of reinforced concrete slabs by Sedaghat Shayegan et al. [21] using the mouth brooding fish algorithm and the optimization of a reinforced concrete ribbed slab by Imran et al. [22] using the genetic algorithm.

Concrete-steel composite floor structures have also been optimized. In the last decade, Kaveh and Shakouri Mahmud Abadi [23] presented the cost optimization of a composite floor using the improved harmony search algorithm, while Omkar et al. [24] presented the multi-objective design optimization of composite structures using the artificial bee colony algorithm. Poitras et al. [25] reported the optimization of composite and steel floor systems using particle swarm optimization. Kaveh and Behnam [26] performed the cost optimization of a composite floor, waffle slab and concrete slab formwork using the charged system search algorithm, while Kaveh and Ahangaran [27] introduced the discrete cost optimization of a composite floor using the social harmony search model. Luo et al. [28] presented an optimal topology design of composite structures using a standard gradient-based search. Žula et al. [29] and Kravanja et al. [30] performed the optimization of a composite floor system using mixed-integer non-linear programming, and Silva and Rodrigues [31] carried out the optimization of steel-concrete composite beams using sequential linear programming. In addition, Jelušič and Kravanja [32] reported the cost optimization of timber-concrete composite floors using multi-parametric mixed-integer non-linear programming, while Kaveh and Ghafari [33] optimized a steel floor system using enhanced colliding body optimization.

In the field of timber floor structures, Stanić et al. [34] and Brank et al. [35] optimized the cross laminated timber plates with ribs. They performed cost optimization of floors using a gradient-based algorithm for mechanical systems by Kegl and Butinar [36]. Mayencourt et al. [37] and Mayencourt and Mueller [38] studied the structural optimization of cross-laminated timber panels. They performed material consumption optimization of structures using the non-linear programming solver (fmincon), implemented in Matlab. Santos et al. [39] applied the cost optimization of cross-insulated timber panels. They also used Matlab software (fmincon solver). Finally, Nesheim et al. [40] reported the cost and ECO2 optimization of timber floor elements for adaptable buildings. For the optimization, they used the mixed-integer sequential linearization procedure.

The above references of timber floor structures do not indicate spans for which timber floor systems are economically viable. There is some information in the literature on the competitive spans of timber floors based on the classical calculations from engineering practice. Kolbitsch [41] reported that a timber floor system with timber beams and sheathing is suitable for spans ranging from 2 to 7 m. While Kolb [42] showed that this system is suitable for spans between 4 and 7.5 m, Cobb [43] proposed this floor system for spans between 1 and 6.5 m. For longer spans, PINE Manufacturers [44] showed that timber floors with glued beams are suitable for spans from 2 to 10 m, while APA [45] reported suitable spans of floors with glued beams from 2.44 m to 15.85 m.

## 2. Materials and Methods

### 2.1. Objectives and Framework of the Study

The novelty of this work is not only the calculation of the optimal design, but also the determination of the competitive spans of timber floor joists, based on the optimization performed and the Eurocode specifications. In contrast to the above-mentioned references in the field of timber floors, which reported individual cost/design optimizations of structures, this work deals with a multi-parametric cost optimization performed for different spans, loads and systems.

With this study, we aimed to verify the above-mentioned spans of timber floor joists and also to increase the knowledge about these spans based on cost optimization and Eurocode standards. In the paper, we propose to determine the competitive spans of two different timber floor systems (see Figure 1):
Floor system A (SYS-A), which contains only timber beams and structural sheathing. However, two types of timber beams are used in SYS-A:
○Sawn wood (SYS-A-SW) and○Glued laminated timber (SYS-A-Gl);Floor system B (SYS-B) made of glued laminated primary timber beams, sawn secondary timber beams and structural sheathing as the main components.


The cost optimizations of the timber floor structures were calculated using mixed-integer non-linear programming (MINLP). MINLP enables simultaneous continuous and discrete optimization. For this purpose, continuous and discrete binary 0–1 variables are defined. The continuous variables are used for optimization of sizes, inner forces, deflections, costs and mass, whilst the discrete variables are employed to calculate material grades and standard sizes. A Modified Outer-Approximation/Equality-Relaxation (OA/ER) algorithm was used to perform optimization of the discrete and non-linear class of the timber floor structure; see Kravanja and Grossmann [46] and Kravanja et al. [47].

We developed an MINLP model for the optimization of timber floor joists in which an objective function of the structure’s self-manufacturing costs is subjected to design, resistance, vibration and deflection constraints. These dimensioning constraints satisfy the conditions of ultimate and serviceability limit states according to Eurocode regulations [48]. Optimal design of the timber floor joists was studied for different important design parameters, such as different floor systems, spans, vertical imposed loads, timber strengths and discrete dimensions of cross-sections. Since a number of individual MINLP optimizations were calculated for the different defined parameters, a multi-parametric MINLP optimization of the floor structure was performed.

Based on the results obtained, a recommended optimal design for timber floor joists was developed. A sensitivity analysis was performed on the contribution of various design parameters to the variability of the overall production costs of timber floor systems, and economically suitable timber floor systems for various spans were found. Finally, competitive spans were determined for the considered timber floor joists.

### 2.2. MINLP Optimization Model

In order to perform a cost optimization of timber floor joists, an MINLP optimization model titled TIMBFJ (TIMBer Floor Joists) was developed. It was modelled in the GAMS environment (General Algebraic Modeling System, commercial software) [49], and it enables the MINLP optimization of a structure for different design parameters such as different loads, spans, strengths and prices. The following assumptions were applied in the optimization model and the calculation:
two different floor systems were considered:
○timber floor with primary beams (SYS-A),○timber floor system with primary and secondary beams (SYS-B);two different sharing systems for mechanically jointed timber floors were used:
○system strength factor *k_sys_*,○Gamma method;depths (heights) of the beams were bounded to be less than or equal to the extreme values which are still, but very rarely, used in the practice:
○30 cm for sawn wood beams,○150 cm for glued laminated timber beams;Eurocode 5 was used to define the dimensioning constraints;prices in Central Europe were considered in the study.

The diversity of different structure systems and joint treatments described above enables us to make an economical comparison between sawn (SYS-A-SW) and glued (SYS-A-Gl) laminated timber beams.

#### 2.2.1. MINLP Model Formulation

The proposed cost and discrete sizing optimization of a timber floor structure is a combined continuous, discrete and non-linear type of optimization. The optimization model is modelled on the following MINLP formulation:


min *z* = *f*(**x**,**y**),
subjected to:  *g_k_* (**x**,**y**) ≤ 0   *k* ∈ *K*
**x** ∈ *X* = {**x** ∈ *R^n^*: **x**^LO^ ≤ **x** ≤ **x**^UP^}
**y** ∈ *Y * = {0,1}^*m*^


MINLP handles continuous variables x and discrete (binary) variables y. Continuous variables are used here to determine the structure costs, loads, loadings, resistances and deflections, while discrete binary variables are employed for choosing the discrete sizes of timber cross-sections. The cost objective function *z* is subject to the design, loading, stress, resistance and deflection (in)equality constraints, *g_k_* (**x**,**y**) ≤ 0, *k* ∈ *K*. These constraints represent the calculation of inner forces and the dimensions of the structure, where the ultimate and serviceability limit state conditions are defined according to Eurocode standards.

The proposed TIMBFJ optimization model includes input data (constants), variables, sets, parameters, the cost objective function of the structure, structural analysis and dimensioning constraints, and logical constraints. The main input data, continuous and discrete variables, sets and parameters are listed in Table 1. While the main input data include a structure span *L* (m), an imposed load *q_k_* (kN/m^2^), various material properties, factors and prices, the continuous variables include the production costs of the structure *COST*, dimensions, cross-section characteristics, stresses, deflections, masses, etc. Sets, discrete variables and parameters are defined to calculate/select the discrete/standard materials (characteristic bending strength of timber) and standard dimensions.

#### 2.2.2. Cost Objective Function

For two different floor systems, SYS-A and SYS-B, the objective function of the self-manufacturing costs COST (EUR/m^2^) of timber floor joists is defined by Equation (1):(1)$$\min:\;COST_{SYS\text{-}A}=C_{M,t}+C_{M,imp}+C_{M,board}+C_{L,board}\;\min:\;COST_{SYS\text{-}B}=C_{M,t}+C_{M,imp}+C_{M,board}+C_{L,board}+C_{M,t,pb}+C_{M,imp,pb}.$$

Different cost items of the objective function for systems SYS-A and SYS-B are presented in Table 2. While floor system SYS-A includes only cost items defined by Equations (2)–(5), system SYS-B also includes items for glulam primary beams determined by Equations (6) and (7). This study considered the cost of spruce wood (Picea Excelsa), one of the most commonly used materials in the construction industry.

In Equations (2)–(7), *c_M,t_* (€/m^3^) is the unit price of the timber, *c_M,imp_* (€/m^3^) is the unit price of timber impregnation, *c_M,board_* (€/m^2^) is the unit price of the prefabricated floor boards and *c_L,board_* (€/m^3^) is the unit price of floor board placing. Index *_pb_* is used to define the dimensions of the glulam primary beam in system SYS-B; see Equations (6) and (7).

On local markets, the unit prices of floor boards depend on their thicknesses. Floor boards with higher thickness are generally more expensive. The unit price of prefabricated timber boards *c_M,board_* = *f*(*d*) is thus included in the cost objective function by applying an approximation function which was formulated by executing multiple regression using MS Excel software with Data Analysis Add-in. The price per unit of timber floor board *c_M,board_* (€/m^2^) is thus calculated to be dependent on the thickness; see Equation (8), where *c_board_* (€/m^2^) stands for the basic price per unit of timber board; the thickness of timber board d must be defined in mm.
(8)$$c_{M,board}=c_{board}\cdot \left(0.0762\cdot d+0.5238\right).$$

#### 2.2.3. Structural Analysis and Dimensioning Constraints

The cost objective function is subjected to structural analysis and dimensioning constraints which are defined in accordance with Eurocode standards [50]; see Equations (9)–(58). Equations (10)–(43) in Table 3 stand for the floor system SYS-A (and for the secondary beams and floor boards of the system SYS B), and Equations (44)–(58) in Table 4 are defined for the glulam primary beams of the floor system SYS-B. In this paper only the most important equations are presented. For a better understanding of the symbols (constants and variables) used in the equations; see Table 1. The timber structure is dimensioned to support its own weight *g_k_* (kN/m^2^, timber beams plus floor boards) and the uniformly distributed variable imposed load *q_k_* (kN/m^2^); see Equation (9):(9)$$\gamma_{g\;}\cdot {\;g}_{k}{+\gamma }_{q\;}\cdot {\;q}_{k}.$$

Partial factors for actions *γ_g_* =1.35 and *γ_q_* =1.5 are defined at the ultimate limit state and *γ_g_* = 1.0 and *γ_q_* = 1.0 at the serviceability limit state.

The ultimate limit state condition, where the maximal design bending stress in timber beam *σ_m,y,d_* (MPa) must not exceed the design bending strength of the timber *f_m,y,d_* (MPa) is calculated by Equation (10). In Equations (11)–(14), the design bending strength is calculated by using a loading sharing factor *k_sys_*. In the equations, *k_h_* is a depth factor, *k_mod_* is a modification factor for duration of load and moisture content, *f_mk_* is the characteristic bending strength of timber, *γ_k_* is the specific weight of timber, *γ_board_* is the specific weight of timber boards and *γ_M_* is a partial factor for material properties. The design bending stress in the above equation is calculated without considering the favorable action of timber boards. Therefore, the Gamma method is also proposed to calculate the effective stiffness of the timber floor system (*EI*)*_eff_*, see Equations (15)–(23). In the equations, *E* and *E_board_* are the modules of elasticity of timber and timber boards, respectively, and *γ*_1_ and *γ*_2_ determine the composite action of the beam and timber board. Equations (17)–(23) define substituted terms *I_c,eff_*, *A_c,eff_*, *a_d,c,eff_*, *a_d,tim_* and *I_tim_* involved in (*EI*)*_eff_*. Constraint Equation (24) determines that the calculated maximal design shear stress in timber beams *τ_max_* (the stress component perpendicular to the grain, MPa) must not exceed the design shear strength of the timber *f_vd_* (MPa). *τ_max_* and *f_vd_* are defined by Equations (25) and (26), respectively, where *f_vk_* (MPa) stands for the shear strength of the timber beam. In the design methodology according to the Gamma method, the maximal design shear stress is expressed by Equation (27).

Equation (28) define the conditions under which the maximal design bending stress in timber board *σ_board,y,d_* (MPa) must not exceed the design bending strength in board *f_board,y,d_* (MPa). *σ_board,y,d_* and *f_board,y,d_* are defined according to Equations (29)–(32). In the equations, *g_k,board_* represents the self-weight (kN/m^2^) of the floor board, depth factor *k_h,board_* is taken to be equal to 1.3 because of the small thickness of the board and *f_board,y,k_* (MPa) represents the characteristic bending strength of timber board. In addition, Equation (33) defines that the maximal design shear stress in the board *τ_board,max_* (the stress component perpendicular to the grain, MPa) must not exceed the design shear strength in the board *f_board,vd_* (MPa). *τ_board,max_* and *f_board,vd_* are determined by Equations (34) and (35), respectively, where *f_board,vk_* (MPa) represents the characteristic shear strength of timber board.

The serviceability limit state condition, where the instantaneous deflection of the timber beam *u_inst_* (m) must be less than or equal to the recommended value, is defined by Equation (36). In the case when the system strength factor *k_sys_* is used, *u_inst_* is defined by Equation (37). In the case of the Gamma method design, the instantaneous deflection of the timber beam is expressed by Equation (38). The instantaneous deflection of the timber boards *u_inst,board_* (m) must also be less than or equal to the allowable value; see Equation (39). Deflection *u_inst,board_* is expressed by Equation (40). The calculated final deflection of the timber beam *u_fin_* (m) must be less than or equal to the recommended value; see Equation (41). In the case of using the factor *k_sys_*, *u_fin_* is defined by Equation (42), where *ψ*_2_ is the factor for the quasi-permanent value of the variable action and *k_def_* is the deformation factor. In this way, the deflection of the structure resulting from the effect of long-term use and moisture is limited. The final deflection *u_fin_*, calculated by the Gamma method, is determined by Equation (43). The final deflection of the timber board *u_fin,board_* (m), which must be less than the recommended value, is verified by Equation (44). The substituted final deflection *u_fin,board_* is expressed by Equation (45).

The constraint Equation (46) defines that the proportion of the point load acting on a single joist *k_dist_* must be greater than or equal to the acceptable value. *k_dist_* is expressed by Equations (47) and (48), where *EI_b_* (Nm^2^/m) is the equivalent board/plate bending stiffness parallel to the beams and *k_strut_* stands for the partial factor for the installed strutting. Equation (49) defines the condition, where the vertical deflection *w* (m) of the joist caused by the concentrated static force *F* = 1.0 kN must not exceed the allowable deflection *a* (m). Terms *w* and *a* are expressed by Equations (50)–(52), where *k_amp_* is the amplification factor that takes into account the shear deflection, and (*EI*)*_joist_* is the stiffness of the joist, which in the case of the Gamma method used is equal to the effective stiffness of the timber floor system (*EI*)*_eff_*, Equation (53).

According to Equations (54)–(56), the fundamental frequency of vibration of a rectangular residential floor *f*_1_ (Hz) should exceed 8 Hz, where *EI_l_* (Nm^2^/m) is the equivalent bending stiffness of the joists divided by the joist spacing in the beam direction and *mass_v_* (kg/m^1^) is the mass of the timber floor structure. Finally, the calculated unit impulse floor velocity ν (1/Ns) must be less than or equal to the allowable floor velocity *ν_allow_* (1/Ns); see Equations (57)–(62), where *n*_40_ represents the number of first-order modes with natural frequencies less than 40 Hz, *b_floor_* (m) represents the floor width, *b_vr_* denotes the velocity response value, *ξ* is the modal damping coefficient, and the other symbols are defined as above.

The above equations are used for the floor system SYS-A and for the upper part of the floor system SYS-B (for the secondary beams and timber boards of the system SYS-B). Equations (63)–(82) are defined for the calculation of the primary glulam beams of the system SYS-B. These equations are nearly identical to those of the system SYS-A, so the terms in the equations are indexed *pb* (primary beam). Equations (63)–(67) define that the design bending stress in the primary beam *σ_m,y,d,pb_* (MPa) must not exceed the design bending strength of the timber of the primary beam *f_m,y,d,pb_* (MPa), Equations (68)–(70) define that the design shear stress in the primary beam *τ_max,pb_* (MPa) must not exceed the design shear strength of the timber of the primary beam *f_vd,pb_* (MPa), Equations (71) and (72) define that the instantaneous deflection of the primary beam *u_inst,pb_* (m) must be less than or equal to the recommended value, Equations (73) and (74) calculate that the final deflection of the primary beam *u_fin,pb_* (m) must be less than or equal to the recommended value, Equations (75)–(76) regulate that the proportion of the point load acting on a single primary beam *k_dist,pb_* must be greater than or equal to the acceptable value, and Equations (77)–(79) limit the vertical deflection of the primary beam *w_pb_* (m) caused by the concentrated static force *F* = 1.0 kN.

Structural analysis and dimensioning (in)equality constraints also determine the timber floor joist dimensions and the material properties to be calculated inside the defined limits. The characteristic shear strength of the timber beams *f_vk_* (MPa), the specific weight of the timber beams *γ_k_* (kN/m^3^) and the modulus of elasticity of the timber *E* (MPa) are substituted in the optimization model as functions of the characteristic bending strength of timber *f_mk_* (MPa); see Equations (80)–(82).
(80)$$f_{vk}=0.4311+0.0858\cdot f_{mk}$$
(81)$$\gamma_{k}=2.3+0.05\cdot f_{mk}$$
(82)$$E=-3714.3+970.24\cdot f_{mk}-14.881\cdot f_{mk}{}^{2}.$$

The glued laminated timber beams are usually manufactured with a height to width ratio smaller than or equal to 12, see Equation (83).
(83)hb≤12.

#### 2.2.4. Logical Constraints

The objective function is subject to logical constraints which determine the discrete values for dimensions of cross-sections and the discrete values for standard materials (strengths). Each discrete dimension *d^st^* (or timber strength) is defined as a scalar product between the vector of *j*, j∈J, discrete dimension alternatives **q_j_** = {*q*_1_, *q*_2_, *q*_3_,*…*, *q_j_*}, and the vector of *j* associated binary variables **y**_j_^st^ = {*y*_1_*^st^*, *y*_2_*^st^*, *y*_3_*^st^*,…, *y_j_^st^*}; see Equation (84). Only one discrete value is selected for the dimension since the sum of the binary variables *y_j_^st^* has to be equal to 1; see Equation (85).
(84)$$d^{st}=\underset{j\in J}{\sum }q_{j}y_{j}^{st}$$
(85)$$\underset{j\in J}{\sum }y_{j}^{st}=1.$$

Figure 2 shows the flowchart of the optimal design of timber floor joists based on multi-parametric MINLP optimization.

### 2.3. Multi-Parametric MINLP Optimization

Multi-parametric MINLP optimization of the production costs of the timber floor joists was performed for combinations of different design parameters:
Nineteen different structure spans: 2 m, 3 m, 4 m, 5 m, 6 m, 7 m, 8 m, 9 m, 10 m, 11 m, 12 m, 13 m, 14 m, 15 m, 16 m, 17 m, 18 m 19 m and 20 m;Five different imposed loads: 1 kN/m^2^, 2 kN/m^2^, 3 kN/m^2^, 4 kN/m^2^ and 5 kN/m^2^;For this purpose, 95 individual MINLP optimizations were executed. For each combination, two different load sharing systems were applied:The system strength factor *k_sys_*;The Gamma method.


In the structural analysis, the following values were assigned: *k_sys_* = 1.1 and *γ*_1_ = 0.2. By assigning a corresponding value *γ*_1_, different types of fastening can be used for the sheathing. The sheathing can be fastened with nails or screws at the appropriate distances as long as the gamma value *γ*_1_ is guaranteed. The gamma factors *γ*_1_ in case the sheathing is fastened with screws or nails can be calculated with the following equations:(86)$$\gamma_{1}={\left(1+\frac{\pi^{2}\cdot E\cdot A\cdot s_{i}}{K_{i}\cdot L^{2}}\right)}^{-1}$$
(87)$$K_{i}=\rho_{m}^{1.5}\cdot \frac{\emptyset }{23};\;\mathrm{for}\;\mathrm{screws}$$
(88)$$K_{i}=\rho_{m}^{1.5}\cdot \frac{\emptyset }{23};\;\mathrm{for}\;\mathrm{nails}\;(\mathrm{without}\;\mathrm{pre}\text{-}\mathrm{drilling}),$$
where *K_i_* is the slip modulus, *ρ_m_* is the mean wood density, *ϕ* is the diameter of the fastener, *s_i_* is the spacing between the fasteners, *L* is the span, and *A* is the cross-sectional area of the sheathing.

Numbers of different discrete alternatives for dimensions of cross-sections and beam spacings were defined; see Table 5. These dimensions (available on the market) were discretized in order to obtain real structural solutions. Only the characteristic bending strength *f_mk_* = 24 MPa is defined in the model because most companies provide just this timber strength. The combinatorics of the defined discrete alternatives in the SYS-A floor system yielded *d_altern_* × *b_altern_* × *h_altern_* × *e_altern_* × *f_mk,altern_* = 3 × 16 × 12 × 12 × 1 = 6912 different structure alternatives (for each different combination between the defined imposed load and structure span). In the SYS-B floor system with primary and secondary timber beams, the number of combinations between different discrete alternatives increased to *d_altern_* × *b_altern_* × *h_altern_* × *e_altern_* × *f_mk,altern_* × *b_pb,altern_*× *h_pb,altern_*× *e_pb,altern_*= 3 × 16 × 12 × 13 × 1 × 12 × 61 × 31 = 169,917,696 different structure alternatives. Only one is optimal (for a given load and span).

The MINLP optimization model TIMBFJ was used. The task of the optimizations was to calculate the minimal self-manufacturing costs of the timber floor joist (€/m^2^), the structure mass (kg/m^2^) and all the dimensions (design variables). The material unit prices (December 2021), labor costs and other data are shown in Table 6. Values from engineering practice in Central Europe were adopted.

The Modified Outer-Approximation/Equality-Relaxation (OA/ER) algorithm by Kravanja and Grossmann [46] and a two-phase MINLP strategy [47] were used. The OA/ER optimization algorithm solves an alternative sequence of non-linear programming (NLP) sub-problems and mixed-integer linear programming (MILP) master problems. The NLP corresponds to the continuous optimization of parameters and yields an upper bound towards the objective to be minimized. The MILP involves a global linear approximation to the superstructure of discrete alternatives in which new discrete sizes are identified. The optimization search is terminated when the predicted lower bound exceeds the upper bound. In cases of non-convex problems, the optimization process is terminated when the NLP solution can no longer be improved. The OA/ER algorithm guarantees global optimality of solutions for convex and quasi-convex optimization problems. The optimization is proposed to be performed sequentially in two different phases in order to accelerate the convergence of the OA/ER algorithm:
The optimization is proposed to start with continuous NLP optimization of the structure (the initialization), where all variables are treated as continuous. The result represents a good starting point and accumulates an effective global linear approximation of the superstructure for the next discrete optimization;The optimization continues with MINLP discrete optimization (a sequence of NLP and MILP solutions) in the second phase until the optimal solution is found.


The optimizations were carried out by the MINLP computer package MIPSYN [50,51]. While GAMS/CONOPT (generalized reduced-gradient method, commercial software) by Drudd [52] was used to solve the NLP sub-problems, GAMS/CPLEX (branch and bound method, commercial software) [53] was employed to find solutions to the MILP master problems.

## 3. Results and Discussion

### 3.1. Recommended Optimal Design for Timber Floor Joists

The recommended optimal design for timber floor joists was determined on the basis of multi-parametric MINLP optimization; see Table 7, Table 8 and Table 9. The recommended design is shown depending on different variable imposed loads *q_k_* and different structure spans *L*.

Comparative analysis of the results shows that it is more economical to increase the timber beam height than to reduce the spacing between beams. If the Gamma method (*γ*_1_ = 0.2) is used instead of the strength system factor (*k_sys_* = 1.1), the costs can be reduced by up to 10%.

Among all the defined conditions relating to timber boards, the serviceability limit state was found to be decisive. This finding applies to all optimal designs of timber floors. The condition of the vertical deflection caused by a concentrated static force *F* = 1.0 kN is decisive for small loads up to 2 kN/m^2^ for all considered spans and all timber floor systems (SYS-A-SW, SYS-A-Gl and SYS-B). In the SYS-A-SW system with spans ranging up to 4 m and for imposed loads larger than 2 kN/m^2^, the main difference between the strength system factor (*k_sys_*) method and the Gamma method (*γ*) is:
If the strength system factor (*k_sys_*) is used, the critical condition is the bending strength of the timbe;While if the Gamma method is used, the critical condition is the final deflection of the timber floor.


In the SYS-A-Gl system, the bending strength of timber was found to be decisive for imposed loads larger than 3 kN/m^2^ for all considered spans. In the SYS-B system, for spans longer than 5 m and imposed loads larger than 2 kN/m^2^, the ultimate limit state is dominant for primary glued lamented timber beams, while the serviceability limit state is dominant for secondary sawn timber beams.

### 3.2. Economically Suitable Spans of Timber Floor Joists

Figure 3 shows diagrams of the calculated optimal self-manufacturing structure costs of nine different spans ranging from 2 m to 10 m. The diagrams were developed for five different variable imposed loads from 1 kN/m^2^ to 5 kN/m^2^, and clearly show that floor joists made of SYS-A-SW sawn timber beams are economically suitable for spans up to 6 m. For these spans, SYS-A-SW is the most appropriate system for all imposed loads. It should be noted that for imposed loads from 1 kN/m^2^ to 3 kN/m^2^ and spans longer than 8 m, nearly the same optimal solutions were calculated, because of the decisive vibration velocity condition.

Figure 4a presents diagrams of the calculated optimal self-manufacturing structure costs of the SYS-A-SW (primary sawn timber beams only) floor system and of the SYS-B (with primary glulam beams and secondary sawn timber beams) floor system. The diagrams are drawn for different spans (from 2 m to 10 m) and for different imposed loads (from 1 kN/m^2^ to 5 kN/m^2^). A comparison between the diagrams clearly shows that the SYS-B system is more economically suitable than the SYS-A-SW system for spans longer than 6 m.

Figure 4b presents diagrams of the calculated optimal self-manufacturing structure costs of the SYS-A-Gl timber floor system (primary glued laminated beams only) and of the SYS-B floor system. For all defined loads, the SYS-A-Gl system is economically suitable for spans up to 10 m, while the SYS-B system is economically suitable for spans up to 20 m. The SYS-B system is more appropriate than SYS-A-Gl for all defined loads and spans.

### 3.3. Calculation Example

In order to interpret the applicability of the comparative diagrams and the recommended optimal design for the timber floor joists, we present a calculation example that determines the most favorable joist structure for the given design parameters and two different load cases.

#### 3.3.1. The structure Self-Weight

In this example, we determine the most competitive design for timber floor joists with a span of *L* = 6 m, subjected to the structure self-weight and the vertical variable imposed load of *q_k_* = 2 kN/m^2^. It should be noted that the Gamma method (*γ*_1_ = 0.2) is applied.

Figure 3b shows that the optimal timber floor SYS-A-SW with a span of 6 m and a vertical imposed load of 2 kN/m^2^ yields self-manufacturing costs of about 42 €/m^2^. The recommended floor joists in Table 7 for a span of 6 m and imposed load of 2 kN/m^2^ are comprised of: thickness of the floor board *d* = 20 mm, width of the timber beam *b* = 60 mm, height of the beam *h* = 300 mm, spacing between floor beams *e* = 0.8 m, characteristic bending strength of timber *f_mk_* = 24 MPa, mass of the structure *mass* = 17.88 kg/m^2^ and self-manufacturing costs of the structure *COST* = 42.44 €/m^2^. The optimal design is shown in Figure 5a. For system SYS-B, Figure 4 shows optimal manufacturing costs for the given design parameter at less than 45 €/m^2^. The recommended design for system SYS-B in Table 9 is comprised of: thickness of the floor board *d* = 20 mm; width of the secondary timber beam *b* = 50 mm; height of the secondary beam *h* = 120 mm; spacing between secondary floor beams *e* = 0.9 m; characteristic bending strength of timber *f_mk_* = 24 MPa; width of the primary glulam beam *b_pb_* = 60 mm; height of the primary glulam beam *h_pb_* = 580 mm; spacing between primary beams *e_pb_* = 2.7 m; mass of the structure *mass* = 16.8 kg/m^2^ and self-manufacturing costs of the structure *COST* = 44.6 €/m^2^; see Figure 5.

#### 3.3.2. The Structure Self-Weight Plus Additional Permanent Load

In the same example (timber floor joists with a span of *L* = 6 m, subjected to the structure self-weight and a vertical variable imposed load of *q_o_* = 2 kN/m^2^), the load on the floor is added, which includes the floor finishing 0.50 kN/m^2^, the insulation between the beams 0.10 kN/m^2^ and the ceiling under the beams 0.30 kN/m^2^, which together gives 0.90 kN/m^2^ of additional permanent load *g_p_*.

Since the comparative diagrams and recommendations are shown in dependence on the vertical imposed load only, the additional permanent load *g_p_* of 0.90 kN/m^2^ must be transformed into a variable load and added to the defined variable imposed load *q_o_* = 2 kN/m^2^. Taking into account Equation (9) and different partial factors for actions at the ultimate limit state for permanent and variable loads, the defined variable imposed load of 2 kN/m^2^ can be modified with the additional permanent load *g_p_* by Equation (89):(89)$$q_{k}=q_{0}+g_{p}\cdot \frac{\gamma_{g}}{\gamma_{q}}=2.0+0.9\frac{1.35}{1.5}=2.81\;\mathrm{kN}/m^{2}.$$

The nearest upper integer value to the calculated modified imposed load *q_k_* is 3 kN/m^2^. Comparing the self-manufacturing costs plotted in Figure 3 and Figure 4 for a span of 6 m and a vertical imposed load of 3 kN/m^2^, the diagram in Figure 3b shows that the cheapest timber floor system is SYS-A-SW at about 44 €/m^2^. The recommended timber structure from Table 7 includes the thickness of floor boards of 20 mm, timber beam width of 50 mm, beam height of 300 mm, beam spacing of 0.6 m, characteristic bending strength of timber 24 MPa, structure mass 18.75 kg/m^2^ and self-manufacturing structure costs of 43.38 €/m^2^, see Figure 6.

## 4. Conclusions

The optimization of timber floor joists takes into account the minimal self-manufacturing costs of a timber structure and the discrete sizes of each element’s cross-sections. Because the optimization problem is discrete and non-linear, mixed-integer non-linear programming (MINLP) was applied to develop the TIMBFJ optimization model, in which an accurate objective function of the structure’s material and labor costs is subjected to design and dimensioning (in)equality constraints, which were defined in accordance with the requirements of Eurocode standards. Two different structural floor systems were considered: a system which contains only timber beams and sheathing, and a system with primary and secondary beams and sheathing (the secondary beams are laid on the primary ones). Two different load sharing systems were applied, a load sharing factor and a system using the Gamma method. Two different materials were used, sawn wood and glued laminated timber. A number of individual MINLP optimizations were calculated for different timber floor systems, vertical imposed loads, spans and alternatives of discrete cross-sections, and a multi-parametric MINLP optimization of the timber floor joists was performed. The optimization problems were solved using the Modified Outer-Approximation/Equality-Relaxation (OA/ER) algorithm and the MINLP computer package MIPSYN.

A comparative analysis of the calculated results shows that it is more economical to increase the timber beam height than to reduce the spacing between beams. If the Gamma method (*γ*_1_ = 0.2) is used instead of the strength system factor (*k_sys_* = 1.1), costs can be reduced by up to 10%. On the basis of the obtained results, a recommended optimal design for timber floor joists was developed. Using these recommendations, an engineer can choose the optimal structure system and material (sawn wood or glulam) for a defined vertical variable imposed load and structure span.

It was found that floor joists made of primary sawn timber beams only (SYS-A-SW) are economically suitable for spans up to 6 m for all defined imposed loads (up to 5 kN/m^2^). This system is the most appropriate floor system for shorter spans. For spans longer than 6 m, a floor system constructed of primary glulam beams and secondary sawn timber beams (SYS-B) is more economically suitable.

For imposed loads up to 5 kN/m^2^, a floor system consisting of primary glulam beams only (SYS-A-Gl) is economically suitable for spans up to 10 m, while a floor system constructed of primary glulam beams and secondary sawn timber beams (SYS-B) is economically suitable for spans up to 20 m. A floor system consisting of primary and secondary beams (SYS-B) is more appropriate than a system consisting of primary glulam beams only (SYS-A-Gl) for all the loads and spans studied. In this way, the competitive spans of the considered timber floor joists were found based on the cost optimization.

## Figures and Tables

**Figure 1 materials-15-03217-f001:**
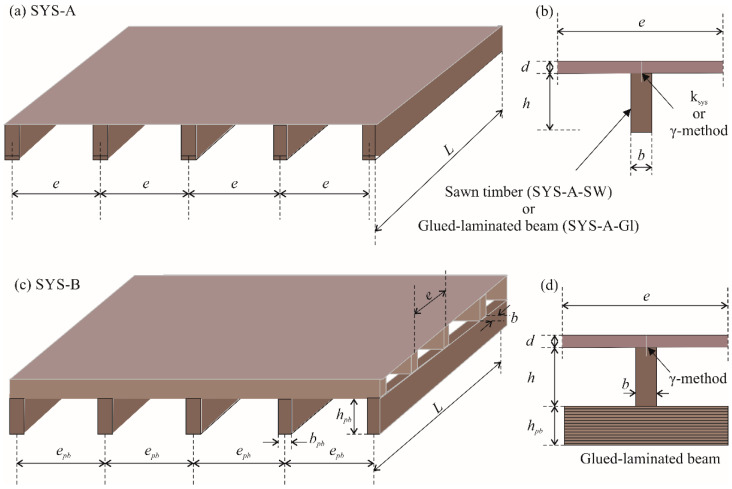
Timber floor joists, (**a**) floor system A (SYS-A), (**b**) vertical cross-section through a timber beam, (**c**) floor system B (SYS-B) with secondary timber beams, (**d**) vertical cross-section through a floor system B.

**Figure 2 materials-15-03217-f002:**
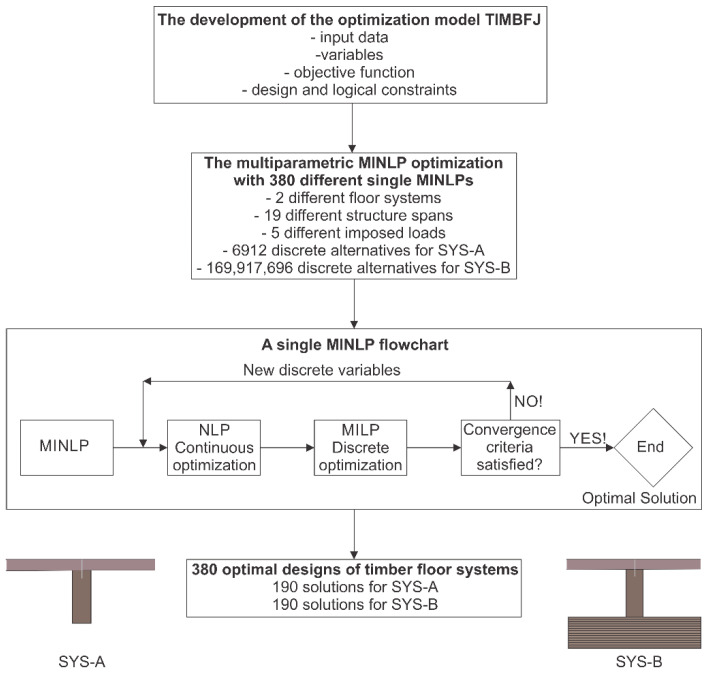
Flowchart of optimal design of timber floor joists based on multi-parametric MINLP optimization.

**Figure 3 materials-15-03217-f003:**
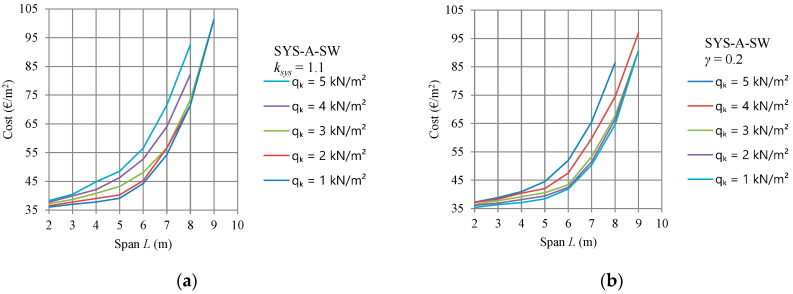
Diagrams of the self-manufacturing costs of the SYS-A-SW timber floor joist system for different spans and imposed loads, (**a**) strength system factor *k_sys_* = 1.1 and (**b**) Gamma method, *γ*_1_ = 0.2.

**Figure 4 materials-15-03217-f004:**
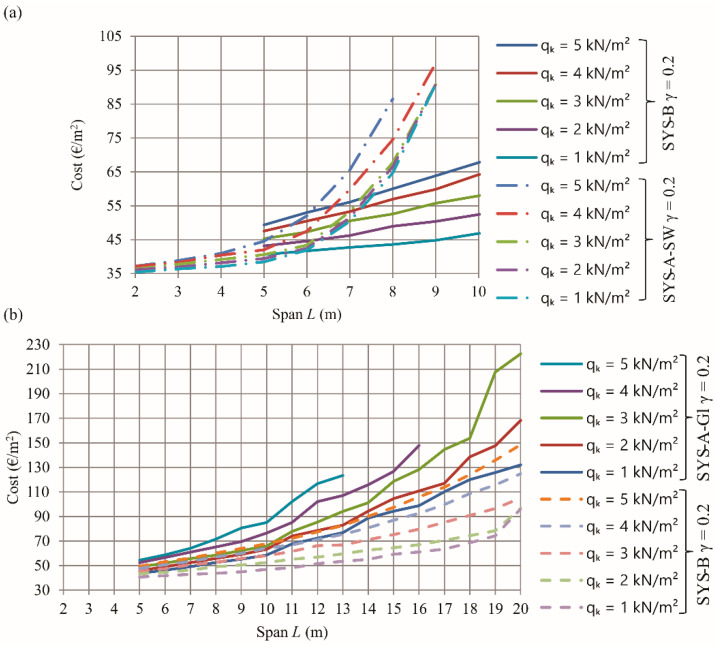
Diagrams of the self-manufacturing costs of timber floor joists for different spans and imposed loads (Gamma method, *γ*_1_ = 0.2), (**a**) comparison between SYS-A-SW and SYS-B systems, (**b**) comparison between SYS-A-Gl and SYS-B systems.

**Figure 5 materials-15-03217-f005:**
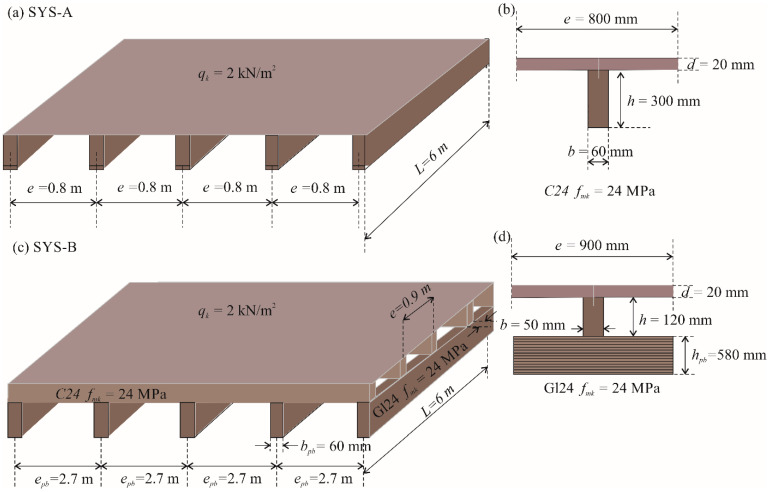
The optimal design of timber floor joists for a structure span *L* = 6 m and variable imposed load *q_k_* = 2 kN/m^2^, (**a**,**b**) floor system A (SYS-A), (**c**,**d**) floor system B (SYS-B).

**Figure 6 materials-15-03217-f006:**
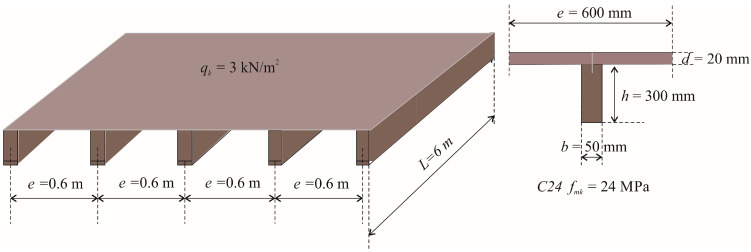
The optimal design of timber floor joists for a structure span *L* = 6 m and variable imposed load *q_k_* = 3 kN/m^2^.

**Table 1 materials-15-03217-t001:** Input data, continuous and discrete variables.

*Input data:*			
*a*	allowable deflection at concentrated static force 1 kN	*g_k_*	self-weigtht of the timber beams plus floor board
*b_floor_*	floor width	*g_k,pb_*	self-weight of the primary beams plus secondary beams plus floor board
*c_board_*	basic price per unit of timber board	*h*	height of the timber beam
*c_L,board_*	price of floor board installation	*h_pb_*	height of the primary timber beam
*c_M,t_*	price of sawn timber or glued laminated timber (C24)	*k_h_*	depth factor
*c_M,imp_*	price of timber impregnation	*k_dist_*	proportion of the point load
*E*	modulus of elasticity of timber	*k_dist,pb_*	proportion of the point load on the primary beam
*E_board_*	modulus of elasticity of the timber boards	*m*	mass of the self-weight of the floor
*f_board,y,k,_*	characteristic bending strength of the timber board	*mass*	mass of the timber floor joist structure
*f_board,vk,_*	characteristic shear strength of the timber board	*mass_v_*	mass of the timber floor structure for vibration frequency calculation
*f_mk_*	characteristic bending strength of timber	*u_fin_*	final deflection of the floor structure
*f_vk_*	characteristic shear strength of timber	*u_fin,pb_*	final deflection of the primary beam
*k_amp_*	amplification factor	*u_fin,board_*	final deflection of the timber board
*k_def_*	deformation factor	*u_inst_*	instantaneous deflection of the floor
*k_mod_*	modification factor to strength values	*u_inst,board_*	instantaneous deflection of the timber board
*k_sys_*	loading sharing factor	*u_inst,pb_*	instantaneous deflection of the primary board
*q_k_*	vertical imposed load	*w*	vertical deflection caused by a concentrated static force 1 kN
*L*	span of floor beams	*w_pb_*	vertical deflection caused by a concentrated static force 1 kN of the primary beam
*γ_board_*	specific weight of timber boards	*COST*	production costs of the structure
*γ_g_*	partial factors for permanent actions	(*EI*)*_b_*	equivalent plate bending stiffness parallel to the beams
*γ_k_*	specific weight of timber beam	(*EI*)*_eff_*	effective stiffness of timber floor system
*γ_M_*	partial factor for material properties	(*EI*)*_jois_*	equivalent bending stiffness of the joists
*γ_g_*	partial factors for imposed actions	(*EI*)_*l*_	equivalent bending stiffness of the joists divided by the joist spacing
*γ* _1_	composite action of the beam and timber board	*M_d_*	design bending moment
*γ* _2_	composite action of the beam	*V_d_*	design shear force
*ξ*	modal damping coefficient	*n* _40_	number of first order modes
*ψ* _2_	factor for the quasi-permanent value of the variable action	ν	floor velocity
		ν*_allow_*	permissible floor velocity
*Variables ***x** ∈ *X*:		*σ_m,y,d_*	design bending stress in timber beam
*b*	width of the timber beam	*σ_m,y,d,pb_*	design bending stress in the primary beam
*b_pb_*	width of the primary timber beam	*τ_board,max_*	design shear stress in timber boards
*b_vr_*	velocity response constant	*τ_max_*	design shear stress in timber beam
*c_board_*	basic price per unit of timber board	*τ_max,pb_*	design shear stress in the primary beam
*d*	thickness of the floor board		
*e*	spacing between floor beams	*Discrete binary variables***y** = {**y**^mat^, **y**^st^}, **y** ∈ *Y*:	
*e_pb_*	spacing between floor primary beams	**y** ^mat^	the sub-vector of the binary variables for bending strengths
*f_m,y,d_*	design bending strength of timber	**y** ^st^	the sub-vector of the binary variables for standard dimensions
*f_m,y,d,pb_*	design bending strength of the timber of the primary beams		
*f_vd_*	design shear strength of timber	*Sets*	
*f_vd,pb_*	design shear strength of the timber of the primary beams	*i, i* ∈ *I*	alternatives of characteristic bending strengths of timber
*f_board,y,d,_*	design bending strength of the timber board	*j, j* ∈ *J*	alternatives of standard dimensions

**Table 2 materials-15-03217-t002:** Cost items of the objective function for floor systems SYS A and SYS B.

Floor System A (SYS A)			Additional Material Cost Items in Timber Floor System B (SYS B)		
Timber:			Timber for primary beams:		
$C_{M,t}=c_{M,t}\cdot h\cdot b/e$		(2)	$C_{M,t,pb}=c_{M,t,pb}\cdot h_{pb}\cdot b_{pb}/e_{pb}$		(6)
Timber impregnation:			Timber impregnation for primary beams:		
$C_{M,imp}=c_{M,imp}\cdot h\cdot b/e$		(3)	$C_{M,imp,pb}=c_{M,imp,pb}\cdot h_{pb}\cdot b_{pb}/e_{pb}$		(7)
Floor boarding:					
$C_{M,board}=c_{M,board}$		(4)			
Floor board placing:					
$C_{L,board}=c_{L,board}$		(5)			
*c_M,t_*	price of timber (€/m^3^)		Where:		
			*b*	width of the timber beam (m)	
*c_M,imp_*	price of timber impregnation (€/m^3^)		*h*	height of the timber beam (m)	
*c_M,board_*	price of the prefabricated floor boards (€/m^2^)		*e*	spacing between timber beams (m)	
*c_L,board_*	price of floor board placing (€/m^3^)		*b_pb_*	width of the timber primary beam (m)	
*c_M,t,pb_*	price of timber for primary beams (€/m^3^)		*h_pb_*	height of the timber primary beam (m)	
*c_M,imp,pb_*	price of timber impregnation primary beams (€/m^3^)		*e_pb_*	spacing between timber primary beams (m)	

**Table 3 materials-15-03217-t003:** Structural analysis and dimensioning constraints (SYS-A).

Ultimate limit state constraints (ULS):	
$\text{-}\;\mathrm{Design}\;\mathrm{bending}\;\mathrm{stress}\;\mathrm{must}\;\mathrm{be}\;\mathrm{smaller}\;\mathrm{than}\;\mathrm{the}\;\mathrm{design}\;\mathrm{bending}\;\mathrm{strength}\;\mathrm{of}\;\mathrm{timber}\;\mathrm{beam}:\;\sigma_{m,y,d}\leq f_{m,y,d}$	(10)
when the loading sharing factor *k_sys_* is used to calculate the design bending strength:	
$\sigma_{m,y,d}=\left[\left(\gamma_{g}\cdot g_{k}+\gamma_{q}\cdot q_{k}\right)\cdot e\cdot L^{2}/8\right]/\left(b\cdot h^{2}/6\right)\;$ $\mathrm{where}\;g_{k}=b\cdot h\cdot \gamma_{k}/e+d\cdot \gamma_{board}\;$	(11),(12)
$f_{m,y,d}=k_{h}\cdot k_{sys}\cdot k_{mod}\cdot f_{mk}/\gamma_{M}\;$ $\;\mathrm{where}\;k_{h}=\min\left[{\left(150/h\right)}^{0.2},1.3\right]$	(13),(14)
when the Gamma method γ is used to calculate the effective stiffness of timber floor system:	
$\sigma_{m,y,d}=\left[0.5\cdot h\cdot E\;\cdot \left(\gamma_{g}\cdot g_{k}+\gamma_{q}\cdot q_{k}\right)\cdot e\cdot L^{2}/8\right]/{\left(EI\right)}_{eff}$	(15)
${\left(EI\right)}_{eff}=E_{board}\cdot I_{c,eff}+\gamma_{1}\cdot E_{board}\cdot A_{c,eff}\cdot a_{d,c,eff}{}^{2}+E\cdot I_{tim}+\gamma_{2}\cdot E_{board}\cdot A_{tim}\cdot a_{d,tim}{}^{2}$	(16)
$\mathrm{where}:\;I_{c,eff}=b_{c,eff}\cdot d^{3}/12$	(17)
$A_{c,eff}=b_{c,eff}\cdot d$ $\;\mathrm{where}\;b_{c,eff}=\min\left(0.2\cdot L,30\cdot d\right)$	(18),(19)
$a_{d,c,eff}=\left(h+d/2\right)-h/2-a_{d,tim}$	(20)
$a_{d,tim}=\gamma_{1}\cdot E_{board}\cdot A_{c,eff}\cdot \left(d+h\right)/\left[2\cdot \left(\gamma_{1}\cdot E_{board}\cdot A_{c,eff}+\gamma_{2}\cdot E\cdot A_{tim}\right)\right]$ $\;\mathrm{where}\;A_{tim}=b\cdot h$	(21),(22)
$I_{tim}=b\cdot h^{3}/12$	(23)
$\text{-}\;\mathrm{Design}\;\mathrm{shear}\;\mathrm{stress}\;\mathrm{in}\;\mathrm{timber}\;\mathrm{beam}\;\mathrm{must}\;\mathrm{not}\;\mathrm{exceed}\;\mathrm{the}\;\mathrm{design}\;\mathrm{shear}\;\mathrm{strength}\;\mathrm{of}\;\mathrm{the}\;\mathrm{timber}:\;\tau_{max}\leq f_{vd}$	(24)
when the loading sharing factor *k_sys_* is used:	
$\tau_{max}=\left[1.5\cdot \left(\gamma_{g}\cdot g_{k}+\gamma_{q}\cdot q_{k}\right)\cdot e\cdot L/2\right]/\left[b\cdot h\right]$	(25)
$f_{vd}=k_{sys}\cdot k_{mod}\cdot f_{vk}/\gamma_{M}$	(26)
when the Gamma method γ is used:	
$\tau_{max}=\left[1.5\cdot \left(\gamma_{g}\cdot g_{k}+\gamma_{q}\cdot q_{k}\right)\cdot e\cdot L/2\cdot 0.5\cdot E\cdot b\cdot {\left(h/2+a_{d,tim}\right)}^{2}\right]/\left[b\cdot {\left(EI\right)}_{eff}\right]$	(27)
$\text{-}\;\mathrm{Design}\;\mathrm{bending}\;\mathrm{stress}\;\mathrm{in}\;\mathrm{timber}\;\mathrm{board}\;\mathrm{must}\;\mathrm{be}\;\mathrm{smaller}\;\mathrm{than}\;\mathrm{the}\;\mathrm{design}\;\mathrm{bending}\;\mathrm{strength}\;\mathrm{of}\;\mathrm{board}:\;\sigma_{board,y,d}\leq f_{board,y,d}$	(28)
$\sigma_{board,y,d}=\left[\left(\gamma_{g}\cdot g_{k,board}+\gamma_{q}\cdot q_{k}\right)\cdot e^{2}/8\right]/\left[1\cdot d^{2}/6\right]$ $\;\mathrm{where}\;g_{k,board}=d\cdot \gamma_{board}$	(29),(30)
$f_{board,y,d}=k_{h,\;board}\cdot k_{mod}\cdot f_{board,y,k}/\gamma_{M}$ $\;\mathrm{where}\;k_{h,board}=1.3$	(31),(32)
$\text{-}\;\mathrm{Design}\;\mathrm{shear}\;\mathrm{stress}\;\mathrm{in}\;\mathrm{timber}\;\mathrm{boards}\;\mathrm{must}\;\mathrm{not}\;\mathrm{exceed}\;\mathrm{the}\;\mathrm{design}\;\mathrm{shear}\;\mathrm{strength}:\;\tau_{board,max}\leq f_{board,vd}$:	(33)
$\tau_{board,max}=\left[1.5\cdot \left(\gamma_{g}\cdot g_{board,k}+\gamma_{q}\cdot q_{k}\right)\cdot e\cdot L/2\right]/\left[1\cdot h\right]$	(34)
$f_{board,vd}=k_{sys}\cdot k_{mod}\cdot f_{board,vk}/\gamma_{M}$	(35)
Serviceability limit state constraints (SLS):	
$\text{-}\;\mathrm{Checking}\;\mathrm{the}\;\mathrm{instantaneous}\;\mathrm{deflection}\;\mathrm{of}\;\mathrm{the}\;\mathrm{timber}\;\mathrm{beam}:\;u_{inst}\;\leq L/300$	(36)
when the loading sharing factor *k_sys_* is used	
$u_{inst}=5\cdot \left(g_{k\;}+q_{k}\right)\cdot e\cdot L^{4}/\left[384\cdot E\cdot b\cdot h^{3}/12\right]+12\cdot \left(g_{k\;}+q_{k}\right)\cdot e\cdot L^{2}/\left[5\cdot E\cdot b\cdot h\right]$	(37)
when *the* Gamma method γ is used	
$u_{inst}=5\cdot \left(g_{k\;}+q_{k}\right)\cdot e\cdot L^{4}/\left[384\cdot {\left(EI\right)}_{eff}\right]+12\cdot \left(g_{k\;}+q_{k}\right)\cdot e\cdot L^{2}/\left[5\cdot E\cdot b\cdot h\right]$	(38)
$\text{-}\;\mathrm{Checking}\;\mathrm{the}\;\mathrm{instantaneous}\;\mathrm{deflection}\;\mathrm{of}\;\mathrm{the}\;\mathrm{timber}\;\mathrm{boards}\;\mathrm{between}\;\mathrm{the}\;\mathrm{timber}\;\mathrm{beams}:\;u_{inst,board}\;\leq e/300$	(39)
$u_{inst,board}=5\cdot \left(g_{k,board\;}+q_{k}\right)\cdot 1\cdot e^{4}/\left[384\cdot E\cdot 1\cdot d^{3}/12\right]+12\cdot \left(g_{k,board\;}+q_{k}\right)\cdot 1\cdot e^{2}/\left[5\cdot E\cdot 1\cdot d\right]$	(40)
$\text{-}\;\mathrm{Checking}\;\mathrm{the}\;\mathrm{final}\;\mathrm{deflection}\;\mathrm{of}\;\mathrm{the}\;\mathrm{timber}\;\mathrm{beam}:\;u_{fin}\;\leq L/250$	(41)
when the loading sharing factor *k_sys_* is used	
$u_{fin}=\left[5\cdot g_{k}\cdot e\cdot L^{4}/\left(384\cdot E\cdot b\cdot h^{3}/12\right)+12\cdot g_{k}\cdot e\cdot L^{2}/\left(5\cdot E\cdot b\cdot h\right)\right]\cdot \left(1+k_{def}\right)+\left[5\cdot q_{k}\cdot e\cdot L^{4}/\left(384\cdot E\cdot b\cdot h^{3}/12\right)+12\cdot q_{k}\cdot e\cdot L^{2}/\left(5\cdot E\cdot b\cdot h\right)\right]\cdot \left(1+\psi_{2}\cdot k_{def}\right)$	(42)
when the Gamma method γ is used	
$u_{fin}=\left[5\cdot g_{k}\cdot e\cdot L^{4}/\left(384\cdot {\left(EI\right)}_{eff}\right)+12\cdot g_{k}\cdot e\cdot L^{2}/\left(5\cdot E\cdot b\cdot h\right)\right]\cdot \left(1+k_{def}\right)+\left[5\cdot q_{k}\cdot e\cdot L^{4}/\left(384\cdot {\left(EI\right)}_{eff}\right)+12\cdot q_{k}\cdot e\cdot L^{2}/\left(5\cdot E\cdot b\cdot h\right)\right]\cdot \left(1+\psi_{2}\cdot k_{def}\right)$	(43)
$\text{-}\;\mathrm{Checking}\;\mathrm{the}\;\mathrm{final}\;\mathrm{deflection}\;\mathrm{of}\;\mathrm{the}\;\mathrm{timber}\;\mathrm{board}\;u_{fin,board}\;\leq e/250$	(44)
$u_{fin,board}=\left[5\cdot g_{k,board}\cdot e\cdot L^{4}/\left(384\cdot E\cdot 1\cdot d^{3}/12\right)+12\cdot g_{k,board}\cdot 1\cdot e^{2}/\left(5\cdot E\cdot 1\cdot d\right)\right]\cdot \left(1+k_{def}\right)+\left[5\cdot q_{k}\cdot 1\cdot e^{4}/\left(384\cdot E\cdot 1\cdot d^{3}/12\right)+12\cdot q_{k}\cdot 1\cdot e^{2}/\left(5\cdot E\cdot 1\cdot d\right)\right]\cdot \left(1+\psi_{2}\cdot k_{def}\right)$	(45)
$\text{-}\;\mathrm{Proportion}\;\mathrm{of}\;\mathrm{the}\;\mathrm{point}\;\mathrm{load}\;\mathrm{acting}\;\mathrm{on}\;a\;\sin\mathrm{gle}\;\mathrm{joist}\;\mathrm{must}\;\mathrm{be}\;\mathrm{greater}\;\mathrm{than}\;\mathrm{the}\;\mathrm{acceptable}\;\mathrm{value}:\;k_{dist}\;\geq 0.3$	(46)
$k_{dist}=k_{strut}\cdot \left(0.38-0.08\cdot ln\cdot \left({\left(EI\right)}_{b}/e^{4}\right)\right)$ $\;\mathrm{where}\;{\left(EI\right)}_{b}=E\cdot 1\cdot d^{3}/12$	(47),(48)
- Vertical deflection caused by concentrated static force F=1.0 kN must not exceed: w ≤a	(49)
$w=1000\cdot k_{dist}\cdot L^{3}\cdot k_{amp}/\left[48\cdot {\left(EI\right)}_{joist}\right]$ $\;\mathrm{where}\;{\left(EI\right)}_{joist}=E\cdot b\cdot h^{3}/12$	(50),(51)
$a=\left(1.8\;for\;L\leq 4000\;\mathrm{mm};\;16,500/L^{1.1};for\;L>4000\;\mathrm{mm}\;\right)$	(52)
$\mathrm{The}\;{\left(EI\right)}_{\mathrm{joist}}\;\mathrm{is}\;\mathrm{modified}\;\mathrm{in}\;\mathrm{Gamma}\;\mathrm{method}\;\mathrm{methodology}\;\mathrm{to}:\;{\left(EI\right)}_{joist}={\left(EI\right)}_{eff}$	(53)
$\text{-}\;\mathrm{Fundamental}\;\mathrm{frequency}\;\mathrm{of}\;\mathrm{vibration}\;\mathrm{of}\;a\;\mathrm{rectangular}\;\mathrm{residential}\;\mathrm{floor}\;\mathrm{should}\;\mathrm{exceed}:\;f_{1}\;\geq 8\;\mathrm{Hz}$	(54)
$f_{1}=\pi /\left(2\cdot L^{2}\right)\sqrt{{\left(EI\right)}_{l}/mass_{v}}$ $\;\mathrm{where}\;mass_{v}=g_{k}\cdot e\cdot 1000/9.81$	(55),(56)
$\text{-}\;\mathrm{Unit}\;\mathrm{impulse}\;\mathrm{floor}\;\mathrm{velocity}\;\mathrm{must}\;\mathrm{be}\;\mathrm{lower}\;\mathrm{than}\;\mathrm{the}\;\mathrm{permissible}\;\mathrm{floor}\;\mathrm{velocity}:\;\nu \leq \nu_{allow}$	(57)
$\nu =4\cdot \left(0.4+0.6\cdot n_{40}\right)/\left(mass_{v}\cdot b_{floor}\cdot L+200\right)$	(58)
$\nu_{allow}=b_{vr}{}^{\left(f_{1}\cdot \xi -1\right)}$	(59)
$\mathrm{where}:\;n_{40}={\left[\left({\left(40/f_{1}\right)}^{2}-1\right)\cdot {\left(b_{floor}/L\right)}^{4}\cdot {\left(EI\right)}_{l}/\left(e\cdot {\left(EI\right)}_{b}\right)\right]}^{0.25}$	(60)
$b_{vr}=180-60\cdot w\;\;\;\;for\;w\leq 1\;\mathrm{mm};\;\;\;\;\;\;\;\;\;\;b_{vr}=180-40\cdot w\;\;\;\;for\;w>1\;\mathrm{mm}\;$	(61),(62)

**Table 4 materials-15-03217-t004:** Structural analysis and dimensioning constraints (SYS-B).

Ultimate limit state constraints (ULS):	
$\text{-}\;\mathrm{Design}\;\mathrm{bending}\;\mathrm{stress}\;\mathrm{must}\;\mathrm{be}\;\mathrm{smaller}\;\mathrm{than}\;\mathrm{the}\;\mathrm{design}\;\mathrm{bending}\;\mathrm{strength}\;\mathrm{of}\;\mathrm{the}\;\mathrm{primary}\;\mathrm{timber}\;\mathrm{beam}:\;\sigma_{m,y,d,pb}\leq f_{m,y,d,pb}$	(63)
$\sigma_{m,y,d,pb}=\left[\left(\gamma_{g}\cdot g_{k,pb}+\gamma_{q}\cdot q_{k}\right)\cdot e_{pb}\cdot L^{2}/8\right]/\left(b_{pb}\cdot h_{pb}{}^{2}/6\right)\;$	(64)
$g_{k,pb}=b\cdot h\cdot \gamma_{k}/e+d\cdot \gamma_{board}+b_{pb}\cdot h_{pb}\cdot \gamma_{k}/e_{pb}\;$	(65)
$f_{m,y,d,pb}=k_{h}\cdot k_{sys}\cdot k_{mod}\cdot f_{mk}/\gamma_{M,pb}$ $\mathrm{where}\;k_{h}=\min\left[{\left(150/h_{pb}\right)}^{0.2},1.3\right]$	(66),(67)
$\text{-}\;\mathrm{Design}\;\mathrm{shear}\;\mathrm{stress}\;\mathrm{in}\;\mathrm{timber}\;\mathrm{beam}\;\mathrm{must}\;\mathrm{not}\;\mathrm{exceed}\;\mathrm{the}\;\mathrm{design}\;\mathrm{shear}\;\mathrm{strength}\;\mathrm{of}\;\mathrm{the}\;\mathrm{primary}\;\mathrm{timber}\;\mathrm{beam}:\;\tau_{max,pb}\leq f_{vd,pb}$	(68)
$\tau_{max,pb}=\left[1.5\cdot \left(\gamma_{g}\cdot g_{k,pb}+\gamma_{q}\cdot q_{k}\right)\cdot e_{pb}\cdot L/2\right]/\left[b_{pb}\cdot h_{pb}\right]$	(69)
$f_{vd,pb}=k_{sys}\cdot k_{mod}\cdot f_{vk}/\gamma_{M}$	(70)
Serviceability limit state constraints (SLS):	
$\text{-}\;\mathrm{Checking}\;\mathrm{the}\;\mathrm{instantaneous}\;\mathrm{deflection}\;\mathrm{of}\;\mathrm{the}\;\mathrm{primary}\;\mathrm{timber}\;\mathrm{beam}:\;u_{inst,pb}\;\leq L/300$	(71)
$u_{inst,pb}=5\cdot \left(g_{k,pb\;}+q_{k}\right)\cdot e_{pb}\cdot L^{4}/\left[384\cdot E_{pb}\cdot b_{pb}\cdot h_{pb}{}^{3}/12\right]+12\left(g_{k,pb\;}+q_{k}\right)\cdot e_{pb}\cdot L^{2}/\left[5\cdot E_{pb}\cdot b_{pb}\cdot h_{pb}\right]$	(72)
$\text{-}\;\mathrm{Checking}\;\mathrm{the}\;\mathrm{net}\;\mathrm{deflection}\;\mathrm{of}\;\mathrm{the}\;\mathrm{primary}\;\mathrm{timber}\;\mathrm{beam}:\;u_{fin,pb}\;\leq L/250$	(73)
$u_{fin,pb}=\left[5\cdot g_{k,pb}\cdot e_{pb}\cdot L^{4}/\left(384\cdot E_{pb}\cdot b_{pb}\cdot h_{pb}{}^{3}/12\right)+12\cdot g_{k,pb}\cdot e_{pb}\cdot L^{2}/\left(5\cdot E_{pb}\cdot b_{pb}\cdot h_{pb}\right)\right]\cdot \left(1+k_{def}\right)+\left[5\cdot q_{k}\cdot e_{pb}\cdot L^{4}/\left(384\cdot E_{pb}\cdot b_{pb}\cdot h_{pb}{}^{3}/12\right)+12\cdot q_{k}\cdot e_{pb}\cdot L^{2}/\left(5\cdot E_{pb}\cdot b_{pb}\cdot h_{pb}\right)\right]\cdot \left(1+\psi_{2}\cdot k_{def}\right)$	(74)
$\text{-}\;\mathrm{Proportion}\;\mathrm{of}\;\mathrm{the}\;\mathrm{point}\;\mathrm{load}\;\mathrm{acting}\;\mathrm{on}\;a\;\sin\mathrm{gle}\;\mathrm{primary}\;\mathrm{beam}\;\mathrm{must}\;\mathrm{be}\;\mathrm{greater}\;\mathrm{than}\;\mathrm{the}\;\mathrm{acceptable}\;\mathrm{value}:\;k_{dist,pb}\;\geq 0.3$	(75)
$k_{dist,pb}=k_{strut}\cdot \left(0.38-0.08\cdot ln\cdot \left({\left(EI\right)}_{eff}/e_{pb}{}^{4}\right)\right)$	(76)
$\text{-}\;\mathrm{Vertical}\;\mathrm{deflection}\;\mathrm{caused}\;\mathrm{by}\;\mathrm{concentrated}\;\mathrm{static}\;\mathrm{force}\;F=1.0\;\mathrm{kN}\;\mathrm{must}\;\mathrm{not}\;\mathrm{exceed}:\;w_{pb}\;\leq a$	(77)
$w_{pb}=1000\cdot k_{dist,pb}\cdot L^{3}\cdot k_{amp}/\left[48\cdot {\left(EI\right)}_{joist,pb}\right]$ $\;\;\;\;\;\;\;\;\;\mathrm{where}\;{\left(EI\right)}_{joist,pb}=E_{pb}\cdot b_{pb}\cdot h_{pb}{}^{3}/12$	(78),(79)

**Table 5 materials-15-03217-t005:** Discrete alternatives of dimensions and strengths.

Variable	Discrete Alternatives
*d* (mm)	20, 25, 30
*b* (mm)	50, 60, 70, 80, 90, 100, 120, 140, 160, 180, 200, 220, 240, 260, 280, 300
*h* (mm)	80, 100, 120, 140, 160, 180, 200, 220, 240, 260, 280, 300
*e* (m)	0.1, 0.2, 0.3, 0.4, 0.5, 0.6, 0.7, 0.8, 0.9, 1.0, 1.1, 1.2
*f_mk_* (MPa)	24
*b_pb_* (mm)	50, 60, 70, 80, 90, 100, 110, 120, 130, 140, 150, 160, 180
*h_pb_* (mm)	300, 320, 340, 360, 380, 400, 420, 440, 460, 480, 500, 520, 540, 560, 580 600, 620, 640, 660, 680, 700, 720, 740, 760, 780, 800, 820, 840, 860, 880, 900, 920, 940, 960, 980, 1000, 1020, 1040, 1060, 1080, 1100, 1120, 1140, 1160, 1180, 1200, 1220, 1260, 1280, 1300, 1320, 1340, 1360, 1380, 1400, 1420, 1440, 1460, 1480, 1500
*e_pb_* (m)	1.0, 1.1, 1.2, 1.3, 1.4, 1.5, 1.6, 1.7, 1.8, 1.9, 2.0, 2.1, 2.2, 2.3, 2.4, 2.5, 2.6, 2.7, 2.8, 2.9, 3.0, 3.1, 3.2, 3.3, 3.4, 3.5, 3.6, 3.7, 3.8, 3.9, 4.0
*e_pb_* (m)	1.0, 1.1, 1.2, 1.3, 1.4, 1.5, 1.6, 1.7, 1.8, 1.9, 2.0, 2.1, 2.2, 2.3, 2.4, 2.5, 2.6, 2.7, 2.8, 2.9, 3.0, 3.1, 3.2, 3.3, 3.4, 3.5, 3.6, 3.7, 3.8, 3.9, 4.0

**Table 6 materials-15-03217-t006:** Material and labor costs and other data.

$c_{M,t}$	Sawn timber or glued laminated timber (C24)	250 €/m^3^ or 500 €/m^3^
$c_{M,imp}$	Timber impregnation	125 €/m^3^
$c_{board}$	Timber boards	21 €/m^2^
$c_{L,board}$	Price of floor board installation	13 €/m^2^
$k_{sys}$	Loading sharing factor	1.1
$k_{mod}$	Modification factor for duration of load and moisture content	0.8
$k_{def}$	Deformation factor	0.8
$\gamma_{M}$	Partial factor for sawn timber or glued laminated timber	1.3 or 1.25
$\gamma_{board}$	Specific weight of timber board	5 kN/m^3^
$\psi_{2}$	Factor for the quasi-permanent value of the variable action	0.3
$\gamma_{g}$	Partial factor for permanent action	1.35
$\gamma_{q}$	Partial factor for variable action	1.50
$k_{strut}$	Partial factor for installed strutting	1.0
$k_{amp}$	Amplification factor to account for shear deflection	1.05
ξ	Modal damping coefficient	0.02
$b_{floor}$	Floor width	10 m
*f_board,y,k_*	Characteristic bending strength of timber board	24 MPa
$\gamma_{1}$	Composite action of the beam and timber board	0.2
$\gamma_{2}$	Composite action of the beam	1.0

**Table 7 materials-15-03217-t007:** Recommended optimal design for timber floor joists constructed of sawn wood in SYS-A-SW (only sawn primary beams).

	Loading Sharing Factor *k_sys_*= 1.1								Gamma Method *γ*_1_ = 0.2							
	Imposed Load *q_k_* = 1 kN/m^2^															
Span *L*:	2 m	3 m	4 m	5 m	6 m	7 m	8 m	9 m	2 m	3 m	4 m	5 m	6 m	7 m	8 m	9 m
*d* (mm)	20	20	20	20	20	20	20	20	20	20	20	20	20	20	20	25
*b* (mm)	50	50	50	50	100	180	300	300	50	50	50	50	70	160	300	260
*h* (mm)	120	160	220	300	300	300	300	300	80	140	180	260	300	300	300	300
*e* (m)	1.1	1.0	1.1	1.1	1.1	1	0.9	0.5	1.1	1.1	1.1	1.1	1	1.1	1.1	0.6
*f_mk_* (MPa)	24	24	24	24	24	24	24	24	24	24	24	24	24	24	24	24
*mass* (kg/m^2^)	11.91	12.80	13.50	14.77	19.55	28.90	45.00	73.00	11.27	12.23	12.86	14.14	17.35	25.27	38.64	58.00
*COST* (€/m^2^)	36.05	37.00	37.75	39.11	44.23	54.25	71.50	101.50	35.36	36.39	37.07	38.43	41.88	50.36	64.68	90.75
	Imposed load *q_k_* = 2 kN/m^2^															
Span *L*:	2 m	3 m	4 m	5 m	6 m	7 m	8 m	9 m	2 m	3 m	4 m	5 m	6 m	7 m	8 m	9 m
*d* (mm)	20	20	20	20	20	20	20	20	20	20	20	20	20	20	20	25
*b* (mm)	50	50	50	50	90	160	300	300	50	50	50	50	60	140	260	260
*h* (mm)	120	180	240	300	300	300	300	300	100	140	200	260	300	300	300	300
*e* (m)	0.9	0.9	0.9	0.9	0.9	0.8	0.9	0.5	0.9	0.9	0.9	0.9	0.8	0.9	0.9	0.6
*f_mk_* (MPa)	24	24	24	24	24	24	24	24	24	24	24	24	24	24	24	24
*mass* (kg/m^2^)	12.33	13.50	14.67	15.83	25.00	31.00	45.00	73.00	11.94	12.72	13.89	15.06	17.88	26.33	40.33	58.00
*COST* (€/m^2^)	36.50	37.75	39.00	40.25	45.25	56.50	71.5	101.5	36.08	36.92	38.17	39.42	42.44	51.50	66.50	90.75
	Imposed load *q_k_* = 3 kN/m^2^															
Span *L*:	2 m	3 m	4 m	5 m	6 m	7 m	8 m	9 m	2 m	3 m	4 m	5 m	6 m	7 m	8 m	9 m
*d* (mm)	20	20	20	20	20	20	20	20	20	20	20	20	20	20	20	25
*b* (mm)	50	50	50	70	100	160	280	300	50	50	50	50	50	120	240	260
*h* (mm)	120	200	260	280	300	300	300	300	100	160	220	280	300	300	300	300
*e* (m)	0.7	0.8	0.8	0.8	0.8	0.8	0.8	0.5	0.8	0.8	0.8	0.8	0.6	0.7	0.8	0.6
*f_mk_* (MPa)	24	24	24	24	24	24	24	24	24	24	24	24	24	24	24	24
*mass* (kg/m^2^)	13.00	14.38	15.69	18.58	23.13	31.00	46.75	73	12.19	13.50	14.81	16.13	18.75	28.00	41.50	58.00
*COST* (€/m^2^)	37.21	38.69	40.09	43.19	48.06	56.50	73.38	101.5	36.34	37.75	39.16	40.56	43.38	53.29	67.75	90.75
	Imposed load *q_k_* = 4 kN/m^2^															
Span *L*:	2 m	3 m	4 m	5 m	6 m	7 m	8 m	9 m	2 m	3 m	4 m	5 m	6 m	7 m	8 m	9 m
*d* (mm)	20	20	20	20	20	20	20	Dimension outside the defined limits	20	20	20	20	20	20	20	20
*b* (mm)	50	50	50	70	100	160	300		50	50	50	50	60	160	180	280
*h* (mm)	140	200	260	280	300	300	300		100	160	240	300	300	300	300	300
*e* (m)	0.7	0.7	0.6	0.6	0.6	0.6	0.7		0.6	0.7	0.7	0.7	0.5	0.7	0.5	0.5
*f_mk_* (MPa)	24	24	24	24	24	24	24		24	24	24	24	24	24	24	24
*mass* (kg/m^2^)	13.50	15.00	17.58	21.43	27.50	38.00	55.00		12.92	14.00	16.00	17.50	22.60	34.00	47.80	68.80
*COST* (€/m^2^)	37.75	39.36	42.13	46.25	52.75	64.00	82.21		37.13	38.29	40.43	42.04	47.50	59.71	74.50	97.00
	Imposed load *q_k_* = 5 kN/m^2^															
Span *L*:	2 m	3 m	4 m	5 m	6 m	7 m	8 m	9 m	2 m	3 m	4 m	5 m	6 m	7 m	8 m	9 m
*d* (mm)	20	20	20	20	20	20	20	Dimension outside the defined limits	20	20	20	20	20	20	20	Dimension outside the defined limits
*b* (mm)	50	50	60	90	120	200	260		50	50	50	50	80	140	280	
*h* (mm)	160	240	240	300	300	300	300		120	180	260	280	300	300	300	
*e* (m)	0.7	0.7	0.5	0.7	0.6	0.6	0.5		0.7	0.7	0.7	0.5	0.5	0.5	0.6	
*f_mk_* (MPa)	24	24	24	24	24	24	24		24	24	24	24	24	24	24	
*mass* (kg/m^2^)	14.00	16.00	20.08	23.50	31.00	45.00	64.60		13.00	14.50	16.50	19.80	26.80	39.40	59.00	
*COST* (€/m^2^)	38.29	40.43	44.80	48.46	56.50	71.50	92.50		37.21	38.82	40.96	44.50	52.00	65.50	86.50	

**Table 8 materials-15-03217-t008:** Recommended optimal design for timber floor joists constructed of glued laminated timber beams in SYS-A-Gl (only glulam primary beams).

	System SYS-A-Gl (Including *γ*_1_ = 0.2)															
	Imposed Load *q_k_* = 1 kN/m^2^															
Span *L*:	5 m	6 m	7 m	8 m	9 m	10 m	11 m	12 m	13 m	14 m	15 m	16 m	17 m	18 m	19 m	20 m
*d* (mm)	20	20	20	20	20	20	20	20	20	20	20	20	25	20	25	25
*b* (mm)	60	60	60	60	60	60	80	80	80	100	100	100	100	120	120	120
*h* (mm)	280	360	440	540	620	720	740	840	940	960	1060	1140	1200	1260	1340	1440
*e* (m)	1.1	1.1	1.1	1.1	1.1	1.1	1.1	1.1	1.1	1.1	1.1	1.1	1.1	1.1	1.2	1.2
*f_mk_* (MPa)	24	24	24	24	24	24	24	24	24	24	24	24	24	24	24	24
*mass* (kg/m^2^)	15.3	16.9	18.4	20.3	21.8	23.7	28.8	31.4	33.9	40.5	43.7	46.3	50.7	58.1	59.4	62.9
*COST* (€/m^2^)	43.5	46.3	49.0	52.4	55.1	58.5	67.6	72.2	76.7	88.5	94.2	98.8	110.2	119.9	125.8	132.0
	Imposed load *q_k_* = 2 kN/m^2^															
Span *L*:	5 m	6 m	7 m	8 m	9 m	10 m	11 m	12 m	13 m	14 m	15 m	16 m	17 m	18 m	19 m	20 m
*d* (mm)	20	20	20	20	20	20	20	20	25	25	25	25	30	30	30	30
*b* (mm)	60	60	60	60	60	60	80	80	80	100	100	100	100	120	120	120
*h* (mm)	280	360	440	520	600	700	720	800	900	920	1000	1100	1180	1180	1300	1420
*e* (m)	0.9	0.9	0.9	0.9	0.9	0.9	0.9	0.9	1.1	1.1	1	1	1.1	1	1	0.9
*f_mk_* (MPa)	24	24	24	24	24	24	24	24	24	24	24	24	24	24	24	24
*mass* (kg/m^2^)	16.5	18.4	20.3	22.1	24.0	26.3	32.4	34.9	35.4	41.8	47.5	51.0	52.5	64.6	69.0	81.3
*COST* (€/m^2^)	45.7	49.0	52.3	55.7	59.0	63.2	74.0	78.4	82.9	94.3	104.5	110.8	117.0	138.5	147.5	168.3
	Imposed load *q_k_* = 3 kN/m^2^															
Span *L*:	5 m	6 m	7 m	8 m	9 m	10 m	11 m	12 m	13 m	14 m	15 m	16 m	17 m	18 m	19 m	20 m
*d* (mm)	20	20	20	20	20	20	20	25	30	30	30	30	30	30	30	30
*b* (mm)	60	60	60	60	60	60	80	80	80	80	80	100	100	100	120	120
*h* (mm)	320	380	460	520	600	680	700	780	880	920	960	1000	1060	1160	1260	1380
*e* (m)	0.8	0.8	0.8	0.8	0.8	0.8	0.8	0.9	1	0.9	0.7	0.8	0.7	0.7	0.6	0.6
*f_mk_* (MPa)	24	24	24	24	24	24	24	24	24	24	24	24	24	24	24	24
*mass* (kg/m^2^)	18.4	20.0	22.1	23.7	25.8	27.9	34.5	36.8	39.6	43.6	53.4	58.8	68.0	73.0	103.2	111.6
*COST* (€/m^2^)	49.0	51.8	55.6	58.4	62.1	65.9	77.8	85.3	94.0	101.1	118.6	128.1	144.6	153.6	207.5	222.5
	Imposed load *q_k_* = 4 kN/m^2^															
Span *L*:	5 m	6 m	7 m	8 m	9 m	10 m	11 m	12 m	13 m	14 m	15 m	16 m	17 m	18 m	19 m	20 m
*d* (mm)	20	20	20	20	20	20	20	25	25	30	30	30	Dimension outside the defined limits	Dimension outside the defined limits	Dimension outside the defined limits	Dimension outside the defined limits
*b* (mm)	60	60	60	60	60	60	60	80	80	80	80	100				
*h* (mm)	340	420	500	580	660	680	680	720	780	920	920	940				
*e* (m)	0.7	0.7	0.7	0.7	0.7	0.6	0.5	0.6	0.6	0.7	0.6	0.6				
*f_mk_* (MPa)	24	24	24	24	24	24	24	24	24	24	24	24				
*mass* (kg/m^2^)	20.2	22.6	25.0	27.4	29.8	33.8	38.6	46.1	48.9	51.8	57.9	69.8				
*COST* (€/m^2^)	52.2	56.5	60.8	65.1	69.4	76.5	85.0	102.0	107.0	115.7	126.7	147.9				
	Imposed load *q_k_* = 5 kN/m^2^															
Span *L*:	5 m	6 m	7 m	8 m	9 m	10 m	11 m	12 m	13 m	14 m	15 m	16 m	17 m	18 m	19 m	20 m
*d* (mm)	20	20	20	20	20	20	25	30	30	Dimension outside the defined limits	Dimension outside the defined limits	Dimension outside the defined limits	Dimension outside the defined limits	Dimension outside the defined limits	Dimension outside the defined limits	Dimension outside the defined limits
b (mm)	60	60	60	60	60	60	80	80	80							
*h* (mm)	380	460	560	600	620	680	720	800	880							
*e* (m)	0.7	0.7	0.7	0.6	0.5	0.5	0.6	0.6	0.6							
*f_mk_* (MPa)	24	24	24	24	24	24	24	24	24							
*mass* (kg/m^2^)	21.4	23.8	26.8	31.0	36.0	38.6	46.1	52.3	56.1							
*COST* (€/m^2^)	54.4	58.6	64.0	71.5	80.5	85.0	102.0	116.7	123.3							

**Table 9 materials-15-03217-t009:** Recommended optimal design for timber floor joists in system SYS-B (glulam primary beams plus sawn secondary beams).

	System SYS-B (Including *γ*_1_ = 0.2)															
	Imposed Load *q_k_*= 1 kN/m^2^															
Span *L*:	5 m	6 m	7 m	8 m	9 m	10 m	11 m	12 m	13 m	14 m	15 m	16 m	17 m	18 m	19 m	20 m
*d* (mm)	20	20	20	20	20	20	20	20	20	20	20	20	20	20	20	20
*b* (mm)	50	50	50	50	50	50	50	50	50	50	50	50	50	50	50	50
*h* (mm)	120	120	140	160	140	180	180	180	180	180	180	180	180	180	180	120
*e* (m)	1.1	1.1	1.1	1.1	1	1.1	1.1	1.1	1.1	1.1	1.1	1.1	1.1	1.1	1.1	1.1
*f_mk_* (MPa)	24	24	24	24	24	24	24	24	24	24	24	24	24	24	24	24
*h_pb_* (mm)	340	420	540	660	720	780	880	920	1040	1140	1180	1280	1400	1440	1480	1500
*b_pb_* (mm)	60	60	60	60	60	80	80	100	100	100	120	120	120	140	160	160
*e_pb_* (m)	2.8	2.8	3.2	3.6	3.3	4	4	4	4	4	4	4	4	4	4	2.5
*mass* (kg/m^2^)	14.5	15.1	15.8	16.4	17.0	18.3	19.0	20.9	22.0	22.8	25.3	26.3	27.6	30.5	33.6	45.5
*COST* (€/m^2^)	40.6	41.7	42.7	43.6	44.8	46.8	48.1	51.4	53.3	54.9	59.2	61.1	63.3	68.6	74.1	96.0
	Imposed load *q_k_* = 2 kN/m^2^															
Span *L*:	5 m	6 m	7 m	8 m	9 m	10 m	11 m	12 m	13 m	14 m	15 m	16 m	17 m	18 m	19 m	20 m
*d* (mm)	20	20	20	20	20	20	20	20	20	20	20	20	20	20	20	20
*b* (mm)	50	50	50	50	550	50	50	50	50	50	50	50	50	50	50	50
*h* (mm)	120	120	120	100	180	140	200	180	160	200	200	180	200	180	140	120
*e* (m)	0.9	0.9	0.9	0.9	0.9	0.9	0.9	0.9	0.9	0.9	0.9	0.9	0.9	0.9	0.8	0.9
*f_mk_* (MPa)	24	24	24	24	24	24	24	24	24	24	24	24	24	24	24	24
*h_pb_* (mm)	480	580	700	720	960	960	1080	1160	1200	1300	1400	1440	1480	1500	1500	1500
*b_pb_* (mm)	60	60	60	60	80	80	100	100	100	120	120	120	140	140	140	160
*e_pb_* (m)	2.7	2.7	2.7	2.1	3.8	3.1	4	3.8	3.4	4	4	3.7	4	3.6	3.2	2.5
*mass* (kg/m^2^)	16.1	16.8	17.8	19.1	20.6	21.4	23.3	24.2	25.5	27.5	28.6	29.8	32.0	33.9	36.0	45.9
*COST* (€/m^2^)	43.2	44.6	46.2	48.9	50.4	52.4	55.0	56.8	59.4	62.5	64.4	66.9	70.5	74.2	78.297	96.5
	Imposed load *q_k_* = 3 kN/m^2^															
Span *L*:	5 m	6 m	7 m	8 m	9 m	10 m	11 m	12 m	13 m	14 m	15 m	16 m	17 m	18 m	19 m	20 m
*d* (mm)	20	20	20	20	20	20	20	20	20	20	20	20	20	20	20	20
*b* (mm)	50	50	50	50	50	50	50	50	50	50	50	50	50	120	50	50
*h* (mm)	140	140	200	160	140	220	160	140	200	180	200	160	140	120	120	120
*e* (m)	0.8	0.8	0.8	0.8	0.8	0.8	0.8	0.8	0.8	0.8	0.8	0.8	0.8	0.8	0.8	0.8
*f_mk_* (MPa)	24	24	24	24	24	24	24	24	24	24	24	24	24	24	24	24
*h_pb_* (mm)	600	720	900	920	960	1200	1180	1200	1420	1440	1500	1480	1450	1480	1500	1500
*b_pb_* (mm)	60	60	80	80	80	100	100	100	120	120	140	140	140	140	140	160
*e_pb_* (m)	2.8	2.7	3.8	3.1	2.6	4	3.1	2.6	3.8	3.3	3.6	3.1	2.7	2.4	2.2	2.2
*mass* (kg/m^2^)	17.6	18.7	21.0	21.8	23.4	25.3	26.8	29.2	30.1	32.3	34.8	36.9	39.9	42.8	46.0	50.8
*COST* (€/m^2^)	45.3	47.3	50.5	52.6	55.7	57.9	61.5	66.1	66.7	70.9	75.1	79.5	85.2	90.8	96.5	105.0
	Imposed load *q_k_* = 4 kN/m^2^															
Span *L*:	5 m	6 m	7 m	8 m	9 m	10 m	11 m	12 m	13 m	14 m	15 m	16 m	17 m	18 m	19 m	20 m
*d* (mm)	20	20	20	20	20	20	20	20	20	20	20	20	20	20	20	20
*b* (mm)	50	50	50	50	50	50	50	50	50	50	50	50	50	50	50	50
*h* (mm)	140	100	160	140	200	160	220	180	200	180	140	120	120	100	100	80
*e* (m)	0.7	0.7	0.7	0.7	0.7	0.7	0.7	0.7	0.7	0.7	0.7	0.6	0.7	0.7	0.7	0.6
*f_mk_* (MPa)	24	24	24	24	24	24	24	24	24	24	24	24	24	24	24	24
*h_pb_* (mm)	680	700	900	960	1180	1200	1400	1420	1480	1480	1460	1500	1500	1480	1500	1500
*b_pb_* (mm)	60	60	80	80	100	100	120	120	140	140	140	140	140	140	160	160
*e_pb_* (m)	2.6	1.9	3.0	2.5	3.6	2.9	3.9	3.3	3.6	3.1	2.6	2.4	2.1	1.8	1.9	1.7
*mass* (kg/m^2^)	19.0	20.2	22.4	24.3	26.5	28.5	30.6	32.6	35.1	37.9	41.0	44.1	48.0	52.8	56.7	61.7
*COST* (€/m^2^)	47.6	50.5	53.3	57.0	59.8	64.1	66.8	71.1	75.3	80.6	86.9	92.5	99.7	108.6	115.6	124.7
	Imposed load *q_k_* = 5 kN/m^2^															
Span *L*:	5 m	6 m	7 m	8 m	9 m	10 m	11 m	12 m	13 m	14 m	15 m	16 m	17 m	18 m	19 m	20 m
*d* (mm)	20	20	20	20	20	20	20	20	20	20	20	20	20	20	20	20
b (mm)	50	50	50	50	50	50	50	50	50	50	50	50	50	50	50	50
*h* (mm)	120	180	160	220	180	240	200	180	180	160	120	120	100	80	80	80
*e* (m)	0.7	0.7	0.7	0.7	0.7	0.7	0.7	0.7	0.7	0.7	0.6	0.6	0.7	0.6	0.7	0.7
*f_mk_* (MPa)	24	24	24	24	24	24	24	24	24	24	24	24	24	24	24	24
*h_pb_* (mm)	680	880	960	1160	1200	1420	1440	1440	1500	1480	1500	1500	1500	1500	1480	1460
*b_pb_* (mm)	60	80	80	100	100	120	120	120	140	140	140	160	140	140	140	160
*e_pb_* (m)	2.1	3.1	2.7	3.6	3.0	3.9	3.3	2.8	3.0	2.5	2.2	2.2	1.7	1.5	1.3	1.3
*mass* (kg/m^2^)	19.8	22.4	24.0	26.8	28.5	31.3	33.3	36.1	39.0	43.0	46.9	51.7	55.7	61.3	67.8	74.9
*COST* (€/m^2^)	49.4	53.0	56.1	60.0	63.8	67.7	72.1	77.4	82.6	90.1	97.4	105.9	113.9	124.0	135.8	148.5

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
