# Peer review of "Optimal Design and Competitive Spans of Timber Floor Joists Based on Multi-Parametric MINLP Optimization"

_materials, 2022, doi:10.3390/ma15093217_

Round 1
Reviewer 1 Report
This manuscript presented optimal design of timber floor joists implementing multi-parameters nonlinear programming. The subject is very interesting. The paper should be polished before its publication from the following angles.
- The language should be carefully checked thoroughly, the typical problems include but not limit to: line 12, 14, 16, 17, 21 etc.
- The paragraph from line 90 to line 95 is suggested to be placed after line 104.
- In Section 2, a flowchart figure is needed to illustrate the programming explicitly.
Author Response
Dear Reviewer #1
Please find attached Responses in Table 1
Thank you very much for your help.
Primož Jelušič

Reviewer 2 Report
Dear author,
Please consider the following comments.
- The equation is part of the text of the article. The equation must be followed by a punctuation mark, such as a comma or period.
- Line 507. It is not recommended to use personal pronouns in a scientific article.
- Line 387. It is not clear what the word "this" means. It is recommended to add a noun connecting the previous and the current sentence after the word "this".
- The article's sections' titles do not comply with the IMRAD format. It is better to use non-IMRAD titles only for subsections, paragraphs, and subparagraphs (with their multi-level numbering). Please rename the sections according to the IMRAD format.
- Are the combined action of the sheathing and floor beams considered when calculating the structures? The effect of combined action should be described in detail.
- It is not clear how the sheathing was fastened to the floor beams.
- It is not clear how the various sheathing's fastening methods influence on stress-strain behavior of the structure. Does this influence study of various fastening methods?
- The species of timber used in the calculation is not indicated. The species of timber greatly affects the strength of the bent elements.
- Did the study look at lumber in a strength class other than C24 graded? If so, why is the dependence on this factor not presented?
- In the "span-cost" diagrams, it is better to approximate the graphs with the derivation of regression equations to calculate these indicators at intermediate loads from 1 to 5 kN/m2.
Author Response
Dear Reviewer #2
Please find attached Responses in Table 2
Thank you very much for your help.
Primož Jelušič

Reviewer 3 Report
Decision:
In this paper, an optimization study based on a mixed-integer nonlinear programming approach (designated as MINLP) has been conducted. The data retrieved provides to do several remarks on manufacturing costs and discrete sizes of the structure for two different timber floor solutions.
In the main, the paper brings no technical novelty but has a large potential to be of interest to engineering practitioners. Nonetheless, the reviewer believes that some minor revisions are still required to make the paper acceptable for publication. Authors should better explain the framework adopted through a flowchart and by addressing the corresponding tools (used software's and refer if are commercial or free); and to improve section 6. Please see the comments given bellow.
- The quality of the English language is satisfactory but some improvements can be made. For instance, authors must avoid long sentences. Sentences can be rephrased or broken down to improve the reading flow. The best example is given in the introduction between lines 63-75. Furthermore, some words are also sometimes repeated in the same paragraph. Check the existence and correct some double spacing between words.
- Tittle :please avoid if possible the use of acronyms in the tittle (MINLP)
- Abstract: line 14: resistance or strength?
- Introduction: Authors are suggested to place the last part of the introduction to an individual section that may present the “Objectives and framework of the study”.
- Introduction is well structured and presents several literature works on optimization methods applied to plates in general. The research opportunity is also addressed well, however it is better highlighted if the objectives of the study are reported in a new section (comment 4). Authors are yet recommended to add the following reference in line 34 to support the load-bearing capacity of timber panels: https://doi.org/10.1016/j.engstruct.2021.112683
- Section 2: Please comment on the availability of the software's that have been used. Are they open-source/free/commercial?
- Section 2, line 157. “Used in the praxis” – used in practice?
- Table 1: it is referred that some variables are continuous. By continuous authors may give the impression that such variables have a probabilistic distribution associated. In the end, authors may just want to highlight that they are real ( ). Please revise.
- Section 2, line 161 & Section 3, line 342. Comment on the date of the estimated costs. Note that such cost evaluation is a very interesting exercise for practitioners, but prices are somehow volatile concerning the current world situation. A note on the date of the reference cost values may be addressed.
- Lines 345-369: Several details are given that could be complemented with the presentation of a flow-chart, in which each task and associated used tool (numerical/analytical).
- Figure 2: Please be consistent and put the ordinates axis uniform for both figure (a) and (b). Also, why not plotting in each plot both methods for a given load? Then, the comparison would be direct.
- Section 6: Figure 4 is similar with Figure 1. Perhaps is better just to address the optimized geometrical parameters used without repeating the whole Figure.
- Section 6: In this section, authors need to address better the type of analysis that has been conducted. It addresses the numerical example, but no numerical model has been described. Please revise this section.
Author Response
Dear Reviewer #3
Please find attached Responses in Table 3.
Thank you very much for your help.
Primož Jelušič
